# Amyloid Beta Peptide (Aβ_1-42_) Reverses the Cholinergic Control of Monocytic IL-1β Release

**DOI:** 10.3390/jcm9092887

**Published:** 2020-09-07

**Authors:** Katrin Richter, Raymond Ogiemwonyi-Schaefer, Sigrid Wilker, Anna I. Chaveiro, Alisa Agné, Matthias Hecker, Martin Reichert, Anca-Laura Amati, Klaus-Dieter Schlüter, Ivan Manzini, Günther Schmalzing, J. Michael McIntosh, Winfried Padberg, Veronika Grau, Andreas Hecker

**Affiliations:** 1Department of General and Thoracic Surgery, Laboratory of Experimental Surgery, Justus-Liebig-University Giessen, German Center for Lung Research, 35392 Giessen, Germany; raymond@rayos.de (R.O.-S.); sigridwilker@web.de (S.W.); Anna.I.Chaveiro@med.uni-giessen.de (A.I.C.); Alisa.Agne@chiru.med.uni-giessen.de (A.A.); Martin.Reichert@chiru.med.uni-giessen.de (M.R.); Anca-Laura.Amati@chiru.med.uni-giessen.de (A.-L.A.); Winfried.Padberg@chiru.med.uni-giessen.de (W.P.); Veronika.Grau@chiru.med.uni-giessen.de (V.G.); 2Department of Internal Medicine, Justus-Liebig-University Giessen, 35392 Giessen, Germany; Matthias.Hecker@innere.med.uni-giessen.de; 3Physiological Institute, Justus-Liebig-University Giessen, 35392 Giessen, Germany; Klaus-Dieter.Schlueter@physiologie.med.uni-giessen.de; 4Department of Animal Physiology and Molecular Biomedicine, Justus-Liebig-University Giessen, 35392 Giessen, Germany; Ivan.Manzini@physzool.bio.uni-giessen.de; 5Institute of Pharmacology and Toxicology, RWTH Aachen University, 52074 Aachen, Germany; gschmalzing@ukaachen.de; 6Department of Biology, University of Utah, Salt Lake City, UT 84112, USA; mcintosh.mike@gmail.com; 7George E. Wahlen Veterans Affairs Medical Center, Salt Lake City, UT 84148, USA; 8Department of Psychiatry, University of Utah, Salt Lake City, UT 84108, USA

**Keywords:** amyloid beta peptide, interleukin-1β, nicotinic acetylcholine receptors, monocytes, systemic inflammation, purinergic signaling, P2X7 receptor, adenosine triphosphate

## Abstract

Amyloid-β peptide (Aβ_1-42_), the cleavage product of the evolutionary highly conserved amyloid precursor protein, presumably plays a pathogenic role in Alzheimer’s disease. Aβ_1-42_ can induce the secretion of the pro-inflammatory cytokine intereukin-1β (IL-1β) in immune cells within and out of the nervous system. Known interaction partners of Aβ_1-42_ are α7 nicotinic acetylcholine receptors (nAChRs). The physiological functions of Aβ_1-42_ are, however, not fully understood. Recently, we identified a cholinergic mechanism that controls monocytic release of IL-1β by canonical and non-canonical agonists of nAChRs containing subunits α7, α9, and/or α10. Here, we tested the hypothesis that Aβ_1-42_ modulates this inhibitory cholinergic mechanism. Lipopolysaccharide-primed monocytic U937 cells and human mononuclear leukocytes were stimulated with the P2X7 receptor agonist 2′(3′)-O-(4-benzoylbenzoyl)adenosine-5′-triphosphate triethylammonium salt (BzATP) in the presence or absence of nAChR agonists and Aβ_1-42_. IL-1β concentrations were measured in the supernatant. Aβ_1-42_ dose-dependently (IC_50_ = 2.54 µM) reversed the inhibitory effect of canonical and non-canonical nicotinic agonists on BzATP-mediated IL-1β-release by monocytic cells, whereas reverse Aβ_42-1_ was ineffective. In conclusion, we discovered a novel pro-inflammatory Aβ_1-42_ function that enables monocytic IL-1β release in the presence of nAChR agonists. These findings provide evidence for a novel physiological function of Aβ_1-42_ in the context of sterile systemic inflammation.

## 1. Introduction

In the context of trauma including major surgical trauma, the pro-inflammatory cytokine interleukin-1 β (IL-1β) is released and importantly contributes to sterile hyperinflammation that may result in barrier dysfunction, translocation of bacteria from the gastro-intestinal tract to the circulation, and, eventually, in sepsis. At the same time, however, IL-1β is needed for an effective prevention and resolution of septic complications [1,2,3,4]. Hence, context-specific mechanisms controlling the production and release of IL-1β are of utmost clinical interest.

Synthesis, maturation, and secretion of IL-1β by monocytic cells are carefully regulated. Pathogen- or danger-associated molecular patterns (PAMPs and DAMPs) are first signals that stimulate pattern recognition receptors expressed, among others, by monocytic cells [3]. An example for a PAMP is lipopolysaccharide (LPS), a cell wall component of Gram-negative bacteria. LPS is a prototypical first signal, that stimulates Toll-like receptor (TLR)-4 and induces the synthesis of the inactive precursor protein pro-IL-1β [4]. Extracellular ATP originating from damaged and/or lytic cells is a typical second DAMP promoting the maturation of pro-IL-1β [2,4]. After binding to the P2X7 receptor (P2X7R), ATP induces the assembly of the NLRP3 (NACHT, LRR, and PYD domains containing protein 3) inflammasome that activates the protease caspase-1. Caspase-1 cleaves pro-IL-1β and finally, mature IL-1β is secreted [2,3,4]. Mechanisms that specifically control the ATP-induced secretion of IL-1β might prevent post-operative hyperinflammation, without fully inhibiting host defense against pathogens that involves several ATP-independent mechanisms of IL-1β release.

Recently, we discovered a novel endogenous cholinergic mechanism that efficiently controls the ATP-mediated NLRP3 inflammasome activation and, hence, release of mature IL-1β by human monocytic cells [5]. We found that activation of nicotinic acetylcholine receptors (nAChRs) containing the evolutionary highly conserved subunits α7, α9, and α10 by classical agonists like acetylcholine (ACh), choline, or nicotine, dose-dependently inhibits the ATP-induced IL-1β release by human monocytic cells [6,7,8]. In this context, phosphocholine (PC) was identified as an agonist of nAChRs [6,7,8]. Interestingly, PC is a ligand of the pentraxin C-reactive protein (CRP), an acute phase protein present in human blood that is a widely used clinical marker for inflammation [9,10]. We found that endogenous CRP-PC complexes are potent nAChR agonists that also inhibit the ATP-dependent inflammasome assembly and IL-1β release by human monocytic cells, whereas PC-free CRP is ineffective [11]. Moreover, we discovered that dipalmitoyl phosphatidylcholine (DPPC) and metabolites of phosphatidylcholines such as lysophosphatidylcholine or glycerophosphocholine (GPC) also function as nAChR agonists [8,12].

Ample evidence suggests that amyloid precursor protein (APP) and its cleavage products interfere with the cholinergic system. The α7 nAChR subunits are direct interaction partners for Aβ peptides [13,14,15,16,17,18,19]. Therefore, we decided to investigate the interaction between APP, nAChRs, and the IL-1β system. APP is an evolutionary highly conserved cell surface protein that is expressed by tissues of all germ layers such as the nervous system, skin, adipose tissue, skeletal muscle, heart, kidney, spleen, gastrointestinal tract, thymus, lung, and liver [20,21,22,23,24]. APP and its cleavage products, including the Aβ peptides, are mainly known for their putative roles in Alzheimer’s disease (AD), the most common dementia and primary neurodegenerative disorder in elderly persons [23,25,26,27,28]. The essential physiological functions of APP that can be predicted because of its evolutionary conservation, however, are incompletely understood.

APP can be cleaved via the non-amyloidogenic pathway that involves the participation of α- and γ-secretases and results in the release of a secreted APP from (sAPP) [21]. In contrast, the amyloidogenic way, that seems to promote the pathogenesis of AD, involves double cleavage by β-secretase (β-site APP-cleaving enzyme1, BACE1) and γ-secretase (a complex containing presenilin-1 or -2) resulting in different types of Aβ, among them Aβ_1-40_ and Aβ_1-42_ [21,22]. Aβ_1-42_ is prone to form fibrils and contributes to plaque formations in the brain of AD patients [21,22,23,29,30,31]. In addition, there are several non-canonical pathways through which APP can be processed as reviewed by Müller and colleagues [32].

Several physiological binding partners of Aβ peptides were identified like apolipoprotein E (ApoE), the receptor for advanced glycosylation end products (RAGE) and serpin–enzyme complex receptor (SEC-R) [33,34]. Moreover, APP can function as a cell surface receptor-like protein that interacts with more than 200 extracellular and intracellular binding partners [32]. In the skin, APP and Aβ peptides seem to regulate epidermal cell differentiation and modulate proliferation and cell migration [20,35]. Intestinal APP and Aβ peptides were suggested to regulate enteric neurons, macrophages, and epithelial cells and even seem to influence absorption and barrier function [20,36]. In addition, multiple non-neuronal immunological effects on leukocytes have been described. Both fibrillary and monomeric Aβ_1-42_ can trigger inflammasome activation in mononuclear leukocytes and induce the secretion of the pro-inflammatory cytokines IL-1β and IL-18 [29,37,38,39,40]. In addition, inflammasome-independent cytokines IL-6, tumor necrosis factor-α (TNF-α), and monocyte chemotactic protein-1 (MCP-1) are released [29,37,39,40,41,42].

In this study, we test the hypothesis that Aβ_1-42_ reverses the inhibitory effect of nicotinic agonists on ATP-mediated release of IL-1β by human monocytic cells. We provide evidence that Aβ_1-42_ functions as an antagonist at monocytic nAChRs containing subunit α7, α9, and α10, in human monocytic U937 cells and primary peripheral mononuclear blood leukocytes (PBMCs). Moreover, we show first clinical data on patients, who underwent major surgery, that support the hypothesis, that this mechanism is also active in vivo.

## 2. Materials and Methods

### 2.1. Reagents

Acetylcholine chloride, nicotine hydrogen tartrate salt, phosphocholine chloride calcium salt tetrahydrate (PC), L-α-glycerophosphorylcholine (GPC), 1,2-dipalmitoyl-SN-glycero-3-phosphocholine (DPPC), C-reactive protein (CRP) isolated from human pleural fluid (Merck Millipore AG732), Aβ_1-42_ (Sigma-Aldrich A9810) as well as reverse Aβ_42-1_ (Sigma-Aldrich SCP0048), and lipopolysaccharide (LPS) from *Escherichia coli* (Sigma-Aldrich L2654) were purchased from Merck (Darmstadt, Germany). Aβ_1-42_ and Aβ_42-1_ were dissolved in an ultrasonic bath and stock solutions were immediately frozen at −20 °C. Later, stock solutions were thawed in an ultrasonic bath for at least 5 min before use. 2′(3′)-O-(4-benzoylbenzoyl)adenosine-5′-triphosphate trieethylammonium salt (BzATP) was provided by Jena Bioscience (Jena, Germany). [V11L, V16D]ArIB and RgIA4 were produced and characterized as described previously [5,7,43,44,45]. Concentrations of all compounds used in this study were optimized in previous experiments on human monocytic U937 cells [6,7,8,11,12].

### 2.2. U937 Cells

Human monocytic U937 cells from a histiocytic lymphoma cell line were obtained from the German Collection of Microorganisms and Cell Culture (Braunschweig, Germany). Cells were cultured at 37 °C, 5% CO_2_ in RPMI 1640 medium (Gibco, Life Technologies, Darmstadt, Germany) with 10% fetal calf serum (FCS, Biochrome, Berlin, Germany) and 2 mM L-glutamine (Gibco). After seeding in a 24-well plate (10^6^ cells/mL), U937 cells were primed for 5 h with 1 µg/mL LPS. Thereafter, cells were stimulated with the P2X7R agonist BzATP (100 µM) for 30 min in the presence or absence of Aβ_1-42_, Aβ_42-1_, or agonists of nAChRs. Cell-free cell culture supernatants were collected and stored at −20 °C.

### 2.3. Human Peripheral Blood Mononuclear Cells (PBMCs)

The study was approved by the ethics committee of the medical faculty Giessen, Germany (No. 90/18) and performed in accordance with the Helsinki Declaration. Each volunteer gave written informed consent. Peripheral blood mononuclear cells (PBMCs) were freshly isolated from blood obtained from self-reported healthy, non-smoking adult volunteers. Blood was drawn into sterile syringes containing 1 mM EDTA (bioWORLD, Dublin, OH, USA) per ml blood and PBMCs were separated on Leucosep gradients (Greiner Bio-One, Frickenhausen, Germany). Before gradient centrifugation, LPS (5 ng/mL) was added to blood samples. Thereafter, PBMCs were cultured in 24-well plates at a density of 1 × 10^6^ cells/mL in Monocyte Attachment Medium (PromoCell, Heidelberg, Germany) for 3 h. Non-adherent cells were removed, and cell culture medium was replaced by fresh RPMI 1640 medium (Sigma-Aldrich R8758). Stimulation with BzATP in the presence or absence of Aβ_1-42_, agonists or antagonists of nAChRs was done as described for U937 cells.

### 2.4. Human Plasma Samples of Patients Undergoing Major Surgery

The study was approved by the ethics committee of the medical faculty Giessen, Germany (No. 159/17) and performed in accordance with the Helsinki Declaration. Written informed consent was given by each patient or patient’s legal representative. Male and female patients aged 48–83 years (median age = 67) undergoing major surgery of the pancreas (*n* = 8; female = 4, male = 4) or esophagectomy (*n* = 4; all male) were recruited at the University Hospital of Giessen, Germany. All patients underwent elective surgery for oncological tumor resection of the esophagus (*n* = 8) or the pancreas (*n* = 4). Patients with preexisting increased inflammatory parameters and septic patients were excluded from the study. The medication prescribed prior to the surgery was checked for potential interactions with IL-1β, as far as known from the literature. All patients underwent anesthesia according to a standardized regimen. The first venous blood sample was collected shortly before the operation as well as 0 to 2 h, 24 to 48 h, 72 to 96 h, and 7 to 9 days after the operation. Venous blood was collected into EDTA-buffered collection tubes (Sarstedt, Nürnberg, Germany) and subsequently subjected to a Ficoll-Hypaque (Sigma-Aldrich, Darmstadt, Germany) gradient. After centrifugation, the upper blood plasma layer was harvested and stored at −20 °C.

### 2.5. Cell Viability and Cytokine Measurements

To test for cell viability at the end of the cell culture experiments on U937 cells and PBMCs, non-radioactive cytotoxicity assay (Promega, Madison, WI, USA) was used to measure lactate dehydrogenase (LDH) activity in cell free supernatants as indicated by the supplier. LDH values are given as percentage of the total LDH content of lysed control cells.

The concentrations of IL-1β were measured in the cell free supernatants and human plasma samples by using the Human IL-1 beta/IL-1F2 DuoSet^®^ enzyme-linked immunosorbent assay (ELISA) from R&D Systems (Minneapolis, MN, USA; sensitivity: less than 1 pg/mL human IL-1β) according to the manufacturer. Aβ_1-42_ concentrations in human plasma samples were measured by the human amyloid beta (aa1-42) Quantikine ELISA Kit from R&D Systems (Minneapolis, MN, USA; sensitivity: 4.73 pg/mL human Aβ_1-42_) as indicated by the manufacturer.

### 2.6. Human P2X7R Expressing HEK293 Cell Line

To generate a stable human P2X7R expressing cell line, a previously cloned cDNA [46] encoding the full-length human P2X7R (hP2X7R) subunit was directionally subcloned via HindIII/ApaI restriction sites into the inducible expression vector pcDNA5/FRT/TO (Invitrogen, Thermo Fisher Scientific, Dreieich, Germany). Except for a H155Y polymorphism, the deduced 595 residues hP2X7R sequence corresponds to UniProt accession ID Q99572.4. The resulting hP2X7R-pcDNA5/FRT/TO plasmid was co-transfected with the Flp recombinase vector, pOG44, into Flp-InTM T-RExTM-293 host cells (Invitrogen) using Lipofectamine LTX reagent (Invitrogen). Stably transfected cells (P2X7R-HEK cells) were selected for hygromycin B resistance in DMEM high glucose medium (Gibco) supplemented with 10% FCS (Biochrome), 2 mM L-glutamine (Gibco), 100 µg/mL hygromycin B (InvivoGen, Toulouse, France), 15 µg/mL blasticidin (InvivoGen), 1 µg/mL % penicillin-streptomycin (Gibco). The Flp-In T-REx system creates isogenic clones, making monoclonal selection unnecessary.

### 2.7. Whole-Cell Patch-Clamp Recordings

For electrophysiological experiments, P2X7R-HEK cells were seeded in cell culture dishes (Nunc, Roskilde, Denmark). P2X7R expression was induced by the addition of 1 μg/mL tetracycline (Sigma-Aldrich) and incubation at 37 °C, 5% CO for 24–48 h. Thereafter, the culture medium was replaced by a bath solution containing 5.4 mM KCl, 120 mM NaCl, 2 mM CaCl_2_, 1 mM MgCl_2_, 10 mM HEPES (4-(2-hydroxyethyl)-piperazine-1-ethanesulfonic acid), and 25 mM glucose (all purchased from Merck; pH 7.4). Whole-cell patch clamp recordings were performed as described previously [6,11]. In brief, patch pipettes were pulled from borosilicate glass capillaries (outer diameter 1.6 mm, Hilgenberg, Malsfeld, Germany) to a resistance of 2 to 4 MΩ using an automated puller (Zeitz, Augsburg, Germany). Pipettes were filled with pipette solution containing 120 mM KCl, 1 mM CaCl_2_, 2 mM MgCl_2_, 10 mM HEPES, 11 mM ethylene glycol tetra acetic acid and 20 mM glucose (all purchased from Merck; pH 7.3). The membrane potential of P2X7R-HEK cells was voltage-clamped to −60 mV and transmembrane currents were amplified with an EPC 9 amplifier (HEKA, Lambrecht, Germany) and acquired via an ITC-16 interface with the Pulse software (HEKA). A pressure-driven microperfusion system was used to apply BzATP (100 µM) and Aβ_1-42_ (5 µg/mL). All experiments were performed at room temperature.

### 2.8. Measurements of Intracellular Ca^2+^

To measure intracellular Ca^2+^ concentrations ([Ca^2+^]_i_), P2X7R-HEK cells were seeded in glass bottom culture dishes (CELLview™, Greiner Bio-One, Kremsmünster, Austria) and P2X7R expression was induced by the addition of 1 μg/mL tetracycline (Sigma-Aldrich T7660) and culturing at 37 °C, 5% CO_2_, for 24–48 h. Thereafter, the culture medium was replaced by the same bath solution as described for the whole-cell patch-clamp recordings. Measurements were performed as described previously [11]. In brief, P2X7R-HEK cells were loaded with 3.3 µM Fura-2/AM (Thermo Fisher Scientific) for 25 min at 37 °C. Fura-2/AM was excited at 340 and 380 nm wavelengths and the fluorescence emission was measured at 510 nm. Three independent batches of P2X7R-HEK cells were used in these experiments and a total number of 411 cells were tracked individually. The fluorescence intensity ratio of 340:380 nm was recorded. All experiments were performed at room temperature.

### 2.9. Statistical Analyses and Data Processing

Results were analyzed using SPSS^®^ (Version 23, IBM^®^, Armonk, NY, USA). The IC_50_ value of Aβ_1-42_ in human U937 cells was determined in GraphPad Prism^®^ (Version 6, GaphPad Software) by fitting log-transformed concentration values and the original effect data. Multiple groups were first analyzed by the non-parametric Kruskal–Wallis test. In case of *p* ≤ 0.05, the non-parametric Mann–Whitney U test was performed to compare between individual groups and again, a *p* ≤ 0.05 was considered as evidence for statistical significance. Paired data were analyzed first by the Friedman test followed by the Wilcoxon signed-rank test. Data were visualized using Inkscape version 0.48.5 r10040 (Free and Open Source Software licensed under the GPL).

## 3. Results

### 3.1. Amyloid Beta Peptide (Aβ_1-42_) Attenuates the Inhibitory Effect of Acetylcholine (ACh) and Nicotine (Nic) on BzATP-Induced Relase of IL-1β by Human Monocytic U937 Cells

Human monocytic U937 cells were primed with 1 µg/mL LPS for 5 h to induce pro-IL-1β synthesis. As expected, LPS-primed U937 cells did not spontaneously release IL-1β, whereas additional application of BzATP (100 µM, 30 min) induced a release of about 50 pg/mL IL-1β in the cell culture supernatant (Figure 1a). ACh and Nic are known canonical agonists of monocytic nAChRs containing subunits α7, α9, and α10 [5]. In accordance with previous studies [6,8], the BzATP-induced release of IL-1β was inhibited in the presence of ACh (7.5 µM; *p* = 0.002; Figure 1a). Interestingly, the inhibitory effect of ACh was dose-dependently reversed by addition of Aβ_1-42_ with an IC_50_ of 2.54 µM; the effect was statistically significant at Aβ_1-42_ concentrations of 5 and 10 µM (*p* = 0.01; Figure 1a). In contrast, reverse Aβ_42-1_ had no impact on the inhibitory effect of ACh. (Figure 1a). Neither Aβ_1-42_ nor Aβ_42-1_ induced IL-1β release from LPS-primed cells (Figure 1b). Moreover, similar results were found for the nAChR agonist Nic (10 µM; Figure 1c). The inhibitory effect of Nic (10 µM) on BzATP-induced IL-1β release (*p* = 0.029) was reversed in the presence of 10 µM Aβ_1-42_ (*p* = 0.029). Cell death was evaluated by measuring LDH and revealed that cell viability was largely unimpaired in these experiments (Appendix A).

### 3.2. Aβ_1-42_ Attenuates the Inhibitory Effect of Non-Canonical Nicotinic Agonists on BzATP-Induced IL-1β Release by Human Monocytic U937 Cells

Recently, we discovered that PC, CRP, GPC, and DPPC function as non-canonical nAChR agonists that inhibit the BzATP-induced release of IL-1β by human monocytic cells via nAChRs containing subunits α7, α9, and/or α10 [6,7,8,12]. Due to the findings that the inhibitory effect of ACh and Nic on BzATP-induced release of IL-1β was attenuated by Aβ_1-42_, we questioned, if this is also true for non-canonical agonists of nAChRs. Similar to ACh (10 µM; *p* = 0.016), the BzATP-induced IL-1β release was inhibited by PC (100 µM; *p* = 0.016), CRP (5 µg/mL; *p* = 0.016), GPC (10 µM; *p* = 0.008) as well as by DPPC (100 µM; *p* = 0.008; Figure 2). Indeed, the inhibitory effect of all non-canonical agonists was blunted in presence on 5 µM Aβ_1-42_ (PC, CRP: *p* = 0.029; GPC, DPPC: *p* = 0.008; Figure 2). Cell death, estimated by measurement of LDH release, remained below 10% in all experiments (Appendix A).

### 3.3. Aβ_1-42_ Attenuates the Inhibitory Effect of PC on BzATP-Induced IL-1β Release by Human PBMCs

Next, we investigated, if the antagonizing effect of Aβ_1-42_ on nAChR agonists is active in primary cells. For this purpose, human PBMCs were freshly isolated and pulsed with 5 ng/mL LPS during the process of PBMC isolation. As expected, the spontaneous secretion of IL-1β by these cells was low (Figure 3a). In response to stimulation with BzATP (100 µM; 30 min) the concentration of IL-1β was increased. This BzATP-induced release of IL-1β was inhibited by PC (200 µM; *p* = 0.018; Figure 3a). In accordance to the findings on U937 cells, 5 µM Aβ_1-42_ reversed the inhibitory effect of PC (*p* = 0.018; Figure 3a). To investigate if PC signals via nAChRs, the conopetide [V11L, V16D]ArIB, a specific antagonist of nAChR containing α7 subunits, and conopeptide RgIA4, specific for nAChRs containing α9 and α10 subunits, were used. [V11L, V16D]ArIB (500 nM; *p* = 0.028) and RgIA4 (50 nM; *p* = 0.018) were applied 10 min prior to PC application and both reversed the inhibitory effect of PC on BzATP-induced release of IL-1β (Figure 3a). In control experiments it was shown that Aβ_1-42_ as well as [V11L, V16D]ArIB and RgIA4 alone had no inhibitory effect on BzATP- or LPS-induced IL-1β concentrations (Figure 3b). Absolute values of IL-1β (pg/mL) obtained in these experiments are given in Appendix A. Cell death, estimated by measurement of LDH release, remained below 8.5% in all experiments (Appendix A).

### 3.4. Aβ_1-42_ Has No Impact on the Ion Channel Activity of the Human P2X7R

Aβ and the ATP-sensitive ionotropic P2X7R are suggested to play an important role in the pathogenesis of Alzheimer’s disease, a neurodegenerative disorder characterized by a sustained inflammatory response [28]. To investigate if Aβ_1-42_ directly changes the ionotropic function of the human P2X7R, we first performed electrophysiological whole-cell patch-clamp measurements on HEK-P2X7R cells (Figure 4). As expected, application of BzATP (100 µM) resulted in repetitive current stimulations due to P2X7R activation (*p* = 0.52; Figure 4a,c). In the next set of experiments, BzATP was first applied alone, which provoked current stimulation (Figure 4b). After washout, the cells were preincubated with Aβ_1-42_ (5 µM) for 50 s, followed by an additional application of BzATP (Figure 4b). Aβ_1-42_ did neither induce changes in ion currents nor impact BzATP-induced current stimulations (*p* = 0.92; Figure 4b,c).

It is known that activation of the P2X7R results in changes of [Ca^2+^]_i_ levels [47]. Therefore, we performed calcium imaging measurements on HEK-P2X7R cells to test for P2X7R ion channel function in the presence or absence of Aβ_1-42_ (Figure 4d,e). As expected, application of BzATP (100 µM) resulted in an increase of [Ca^2+^]_i_. Application of 5 µM Aβ_1-42_ did not cause alterations in [Ca^2+^]_i_ whereas additional application of BzATP induced a rise in [Ca^2+^]_i_. The BzATP-induced peak [Ca^2+^]_i_ changes in the absence (d) or presence (e) of Aβ_1-42_ were not significantly different (*p* = 0.92; Figure 4d,e).

### 3.5. Aβ_1-42_ and IL-1β Concentrations in Human Plasma of Patients Undergoing Major Surgery

Finally, we investigated perioperative IL-1β and Aβ_1-42_ levels in the blood plasma of 12 patients, who underwent major abdominal surgery. Blood was drawn shortly before the operation (*n* = 12), 0 to 2 h (*n* = 10), 24 to 48 h (*n* = 12), 72 to 96 h (*n* = 12), and 7 to 9 days (*n* = 8) after the operation, and the data of all samples were investigated collectively. In most patient samples IL-1β remained below the threshold of detection, and a median of 19.70 pg/mL Aβ_1-42_ was detected (Figure 5). However, in plasma samples with detectable IL-1β levels, significantly (*p* ≤ 0.001) higher Aβ_1-42_ levels were measured (median 56.60 pg/mL) compared to those samples, in which IL-1β remained below the threshold of detection (Figure 5).

## 4. Discussion

APP and its Aβ peptides as well as nAChRs are evolutionary highly conserved and evolved long before the emergence of a nervous system [48,49,50,51], suggesting that their original functions need to be sought in more primitive and ubiquitous systems, such as innate immunity. Most research has investigated the pathophysiological effects of the interaction of Aβ with α7 nAChRs with a special focus on the etiology of AD [52]. In this study, we provide evidence that Aβ_1-42_ functions as an antagonist of monocytic nAChRs containing subunits α7, α9, and/or α10, in human monocytic U937 cells and primary human PBMCs (Figure 6). Thus, we have discovered a novel pro-inflammatory function of Aβ_1-42_ that enables monocytic IL-1β release in the presence of nAChR agonists. In addition, we provide first clinical hints that this mechanism might be functional in vivo.

Our data suggest that Aβ_1-42_ can functionally interact with monocytic nAChRs. Aβ_1-42_ reverses the inhibitory effect of nicotinic agonists on ATP-mediated release of IL-1β in monocytic U937 cells. In previous studies, we found that beside classical nAChR agonists like ACh, choline, and nicotine, non-canonical agonist like PC, endogenous CRP, GPC, and DPPC can activate monocytic nAChRs containing subunits α9, α7, and/or α10 to induce this anti-inflammatory effect [6,7,8,11,12]. Here, we show that Aβ_1-42_ dose-dependently antagonizes the inhibitory effect of classical as well as of non-canonical nicotinic agonists on the BzATP-mediated IL-1β release by human monocytic U937 cells.

Because of the well-known limitations of cell lines, we included primary PBMCs in this study that were freshly isolated from the blood of healthy volunteers. In these experiments, we focused on PC, the lead compound of the non-canonical agonists of monocytic nAChRs [5,7,8] and directly compared the effects of conopeptides [V11L, V16D]ArIB and RgIA4 to the effects of Aβ_1-42_. We confirmed that, also in human PBMCs, PC signals via subunits α7, α9, and/or α10 to inhibit the BzATP-induced release of IL-1β [6,7]. The effect of Aβ_1-42_, again, strikingly resembles those of the conopeptides [V11L, V16D]ArIB (α7-specific) and RgIA4 (α9/α10-specific).

In our experimental setting, we found that Aβ_1-42_ antagonizes the inhibitory effects of nicotinic agonists with an IC_50_ of 2.54 µM, corresponding to about 10 µg/mL. Whether the Aβ_1-42_ concentrations used in the present study are physiological for monocytic cells is difficult to decide. Roher and colleagues compared plasma Aβ levels in AD patients and in age-matched control subjects at four time points over a period of 12 months [53]. Aβ_1-40_ and Aβ_1-42_ plasma levels fluctuated widely among the individuals in both AD subjects and controls with overall mean values of 384 pg/mL and 132 pg/mL, respectively [53]. In the light of these data, an IC_50_ of about 10 µg/mL seems to be extremely high. However, APP is expressed by vascular endothelial cells [50]. Moreover, activated platelets and leukocytes can secrete APP and its Aβ peptides, thus contributing to 90% of peripheral serum concentrations [50,54,55]. As monocytes closely interact with endothelial cells and platelets [56,57] and may secrete Aβ_1-42_ themselves, Aβ_1-42_ might function as a paracrine and/or autocrine factor and local concentrations possibly reach micromolar concentrations.

How does Aβ_1-42_ reverse the inhibitory effects of nicotinic agonists on the ATP-induced release of IL-1β? To tackle the question, whether Aβ_1-42_ directly modulates ATP receptor functions, we made use of HEK293 cells overexpressing the human P2X7R. Upon stimulation with BzATP, we observed the expected changes in ion currents and cytoplasmic Ca^2+^ concentrations, which remained unchanged in the presence of Aβ_1-42_. Therefore, in the present study Aβ_1-42_ seems not to modulate P2X7R ion channel activity. Moreover, this result favors the hypothesis that Aβ_1-42_ interacts with monocytic nAChRs. However, some studies pointed out an important role of the P2X7R in Aβ signaling [28,58]. These controversial results might be due to the different experimental protocols and cells used here (monocytic cells) and in the previous studies (microglia).

Ample evidence suggests that Aβ_1-42_ directly interacts with conventional nAChRs [59,60,61,62]. Conventional nAChRs of the neuronal type are ligand-gated ion channels composed of α and β subunits that form heteromeric or homomeric pentamers with a conductance for positively charged ions like Na^+^, K^+^, and Ca^2+^ [63,64]. Homomeric α7 nAChRs and heteromeric α4β2 are the two most abundant types of nAChRs in the human central nervous system [65]. Wang and colleagues were the first to identify an interaction of the α7 nAChRs and Aβ_1-42_ in the brain [59,60]. Later, it was shown that α4β2, α4α5β2, and α2αβ nAChRs can also interact with APP and its Aβ peptides [61,62]. In contrast to our data, however, the affinity of Aβ_1-42_ to α7 nAChRs in the brain is in the low picomolar range [51,59].

A direct modulation of the ion channel functions of α7 nAChR by Aβ_1-42_ was first shown by Petti and colleagues in brain slices, where Aβ_1-42_ inhibited postsynaptic nAChR-mediated currents [15,66]. Seemingly, the Aβ_1-42_ amino acid residues 1–28 are responsible for this interaction [60,66]. The literature about the consequences of Aβ peptide binding to nAChRs is, however, contradictory [52]. Both, agonistic and antagonistic effects of Aβ peptides on the α7 nAChR have been described [15,16,51,52,67,68,69,70,71]. Moreover, we and others demonstrated that leukocytic nAChRs do not function as classical ion channels, but rather exert metabotropic functions [5,6,7,72,73,74,75,76,77,78]. Interestingly, ion channel activity of nAChRs was detected in human monocyte-derived macrophages [79] and macrophages derived from monocytic THP-1 cells [80] in the presence of the positive allosteric modulator PNU-120596. However, as the structure and function of monocytic nAChRs still remain to be elucidated, we can only speculate on their interaction with Aβ_1-42_. Due to the lack of specific antibodies and the low expression of nAChRs by leukocytes, it is extremely difficult to provide evidence for their direct interaction with Aβ_1-42_.

Numerous studies have shown pro-inflammatory functions of Aβ. Treatment of PBMCs with Aβ_1-42_ leads to the secretion of pro-inflammatory cytokines like IL-1β, IL-6, IL-18, monocyte chemotactic protein-1 (MCP-1), and TNF-α [37,39,40,42,81,82]. Fibrillar Aβ_1-42_ is supposed to directly interact with TLR2 and TLR4 to trigger TNF-α secretion in a monocytic cell line [83]. TLR activation is known to induce the expression of IL-1β and of components of the inflammasome. In our experiments, however, Aβ_1-42_ was applied together with BzATP for only 30 min, which is probably not enough time for an efficient induction of gene expression and protein synthesis. Other studies found that fibrillar and monomeric Aβ_1-42_ can activate the NLRP3 inflammasome, thereby inducing the release of mature IL-1β [29,38,84,85]. This probably depends on the internalization of Aβ_1-42_ fibrils via the scavenger receptor CD36, followed by lysosomal rupture, a known signal of NLRP3 assembly [29,86,87]. In the present study, we used soluble Aβ_1-42_ and did not see a significant release of IL-1β in the absence of BzATP, and the BzATP-induced release of IL-1β was not changed in presence of Aβ_1-42_. Therefore, it is unlikely that Aβ_1-42_ boosted the expression of pro-IL-1β or its cleavage machinery or directly activated the NLRP3 inflammasome in our experimental setting.

To obtain a first indication whether IL-1β and Aβ_1-42_ blood levels are linked in vivo, blood plasma samples from patients, who underwent major surgery, were investigated. We did not perform a correlation analysis because IL-1β was undetectable in a large proportion of the samples. Systemic IL-1β levels are known to be low, even during systemic inflammation, due to its short half-life in the circulation [88,89]. Instead, we compared Aβ_1-42_ levels in samples with undetectable versus detectable IL-1β levels. In plasma samples with detectable IL-1β, Aβ_1-42_ levels were indeed higher compared to samples, in which IL-1β was undetectable. This is in line with our hypothesis that Aβ_1-42_ fosters the release of IL-1β. However, these data should be interpreted with utmost caution and further careful clinical research is warranted, because we only investigated a small cohort of patients, pooled data from blood collections at different perioperative time points and do not prove causality at all. Furthermore, patient Aβ_1-42_ levels measured in our study were about half as low compared to those reported for healthy blood donors [53]. We assume that this is due to technical reasons, as we separated plasma from blood cells on a density gradient that certainly resulted in dilution. Even the highest Aβ_1-42_ concentration measured in patient blood plasma, which is in the range of 100 pg/mL, is far below the Aβ_1-42_ concentrations needed to inhibit ATP-induced IL-1β release in monocytic U937 cells and human PBMCs (IC_50_ = 10 µg/mL). Hence, Aβ_1-42_ is probably not active systemically, but might reach sufficient concentrations in the micro-milieu of traumatized tissues, as inflammation is known to upregulate APP and its cleavage machinery resulting in increased Aβ levels [90,91].

## 5. Conclusions

We conclude from our study that Aβ_1-42_ enables the ATP-induced release of monocytic IL-1β despite the presence of potent nicotinic agonists. Although we do not know if Aβ_1-42_ directly interacts with leukocytic nAChRs, Aβ_1-42_ most probably functions as a nicotinic antagonist. As enabling an ATP-induced IL-1β release can prevent infection and fatal sepsis, we believe that we discovered one of the vital functions of APP and its peptides that contributed to their striking conservation during evolution.

Our findings are certainly of clinical interest, especially in the context of surgery-induced inflammation. High Aβ_1-42_ concentrations in patient blood might predict inflammatory complications in response to major surgery and our findings might pave the way towards new therapeutic avenues. However, more studies including experimental in vivo studies and extended carefully controlled clinical studies are warranted to estimate the clinical relevance of our results.

## Figures and Tables

**Figure 1 jcm-09-02887-f001:**
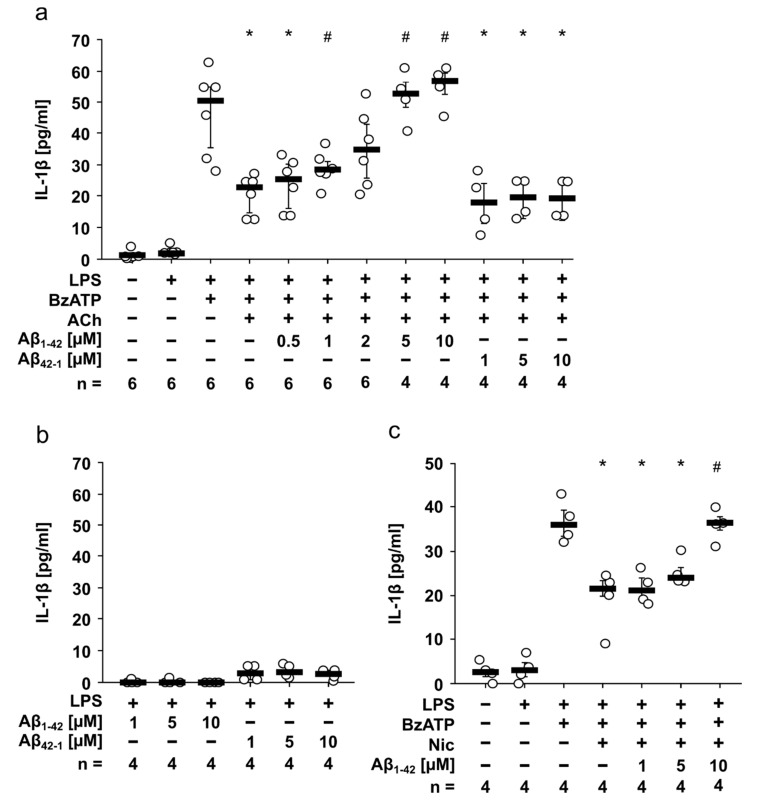
Amyloid beta peptide (Aβ_1-42_) reverses the inhibitory effect of acetylcholine (ACh) and nicotine (Nic) on BzATP-induced interleukin-1β (IL-1β) release by human monocytic U937 cells. U937 cells were primed with lipopolysaccharide (LPS) from *Escherichia coli* for 5 h. Thereafter, the P2X7 receptor agonist 2(3)-O-(4-benzoylbenzoyl)adenosine-5-triphosphate (BzATP; 100 µM) was added for another 30 min and IL-1β was measured by ELISA in cell culture supernatants. (**a**) The nicotinic agonist ACh (7.5 µM) inhibited the BzATP-induced release of IL-1β. This inhibitory effect was dose dependently blunted by Aβ_1-42_ with an IC_50_ = 2.54 µM. Reverse Aβ_42-1_ (1–10 µM) had no impact. (**b**) In control experiments, neither Aβ_1-42_ nor Aβ_42-1_ induced changes in IL-1β release by LPS-primed U937 cells. (**c**) The inhibitory effect of Nic (10 µM) was blunted in presence of 10 µM Aβ_1-42_. All data are shown as individual data points, bar represents median, whiskers encompass the 25th to 75th percentile. Statistical analyses were performed using the Kruskal–Wallis followed by Mann–Whitney U test. * *p* ≤ 0.05, different from LPS-primed cells stimulated with BzATP alone. # *p* ≤ 0.05, different from LPS-primed cells stimulated with BzATP and the corresponding nicotinic agonist without Aβ_1-42_.

**Figure 2 jcm-09-02887-f002:**
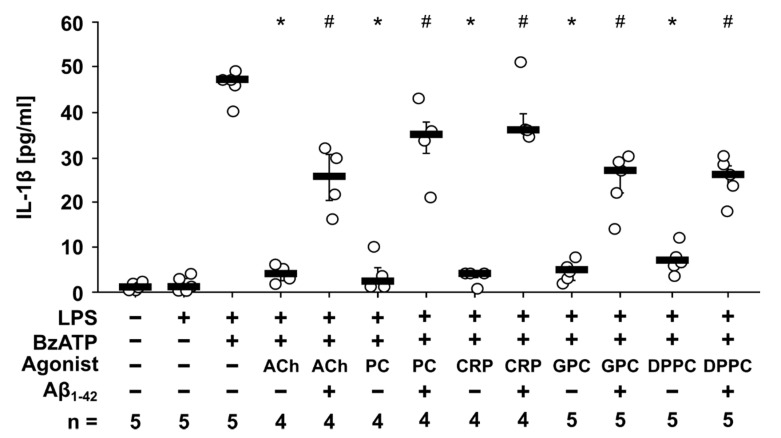
Amyloid beta peptide (Aβ_1-42_) blunts the inhibitory effect of non-canonical nicotinic agonists on BzATP-induced interleukin-1β (IL-1β) release by human monocytic U937 cells. U937 cells were primed with lipopolysaccharide (LPS) from *Escherichia coli* for 5 h. Thereafter, the P2X7 receptor agonist BzATP (100 µM) was added for another 30 min in the presence or absence of Aβ_1-42_ (5 µM) and different nicotinic agonists: acetylcholine (ACh; 10 µM), phosphocholine (PC; 100 µM), C-reactive protein (CRP; 5 µg/mL), glycerophosphorylcholine (GPC; 10 µM) and dipalmitoylphosphatidylcholine (DPPC; 100 µM). BzATP-induced release of IL-1β was measured by ELISA in cell culture supernatants. All data are shown as individual data points, bar represents median, whiskers encompass the 25th to 75th percentile. Statistical analyses were performed using the Kruskal–Wallis followed by Mann–Whitney U test. * *p* ≤ 0.05, different from LPS-primed cells stimulated with BzATP alone. # *p* ≤ 0.05, different from LPS-primed cells stimulated with BzATP and the corresponding nicotinic agonist without Aβ_1-42_.

**Figure 3 jcm-09-02887-f003:**
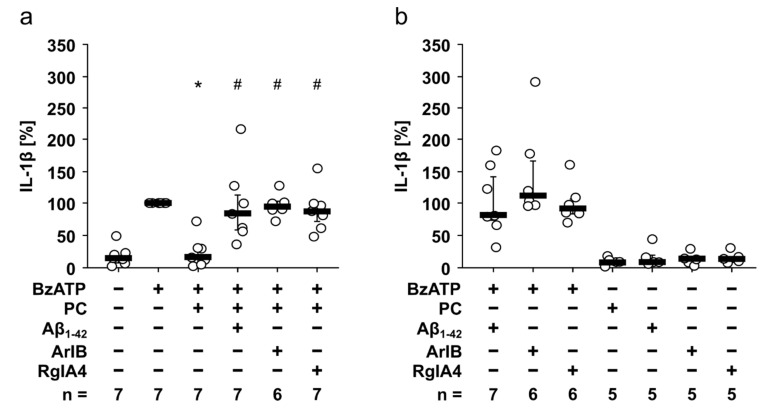
Amyloid beta peptide (Aβ_1-42_) attenuates the inhibitory effect of phosphocholine (PC) on the BzATP-induced interleukin-1β (IL-1β) release by human peripheral blood mononuclear cells (PBMCs). PBMCs freshly isolated from healthy human volunteers were pulsed with lipopolysaccharide (LPS; 5 ng/mL) during the process of PBMC isolation and cultured for 3 h. The cells were stimulated with the P2X7 receptor agonist BzATP (100 µM) in the presence or absence of phosphocholine (PC; 200 µM), Aβ_1-42_ (5 µM), [V11L, V16D]ArIB (ArIB; 500 nM) and/or RgIA4 (50 nM) for 30 min. BzATP-induced release of IL-1β was measured by ELISA in cell culture supernatants. The IL-1β concentration in experiments, in which PBMCs were stimulated with BzATP alone, was set to 100% and all other values were calculated accordingly. (**a**) The inhibitory effect of PC was reversed in the presence of Aβ_1-42_, RgIA4 as well as ArIB. (**b**) In control experiments, Aβ_1-42_, RgIA4, and ArIB had no impact on BzATP- and LPS-induced IL-1β release. All data are shown as individual data points, bar represents median, whiskers encompass the 25th to 75th percentile. Statistical analyses were performed using the Friedman test followed by the Wilcoxon signed-rank test. * *p* ≤ 0.05, different from LPS-primed cells stimulated with BzATP alone. # *p* ≤ 0.05, different from LPS-primed cells stimulated with BzATP and PC.

**Figure 4 jcm-09-02887-f004:**
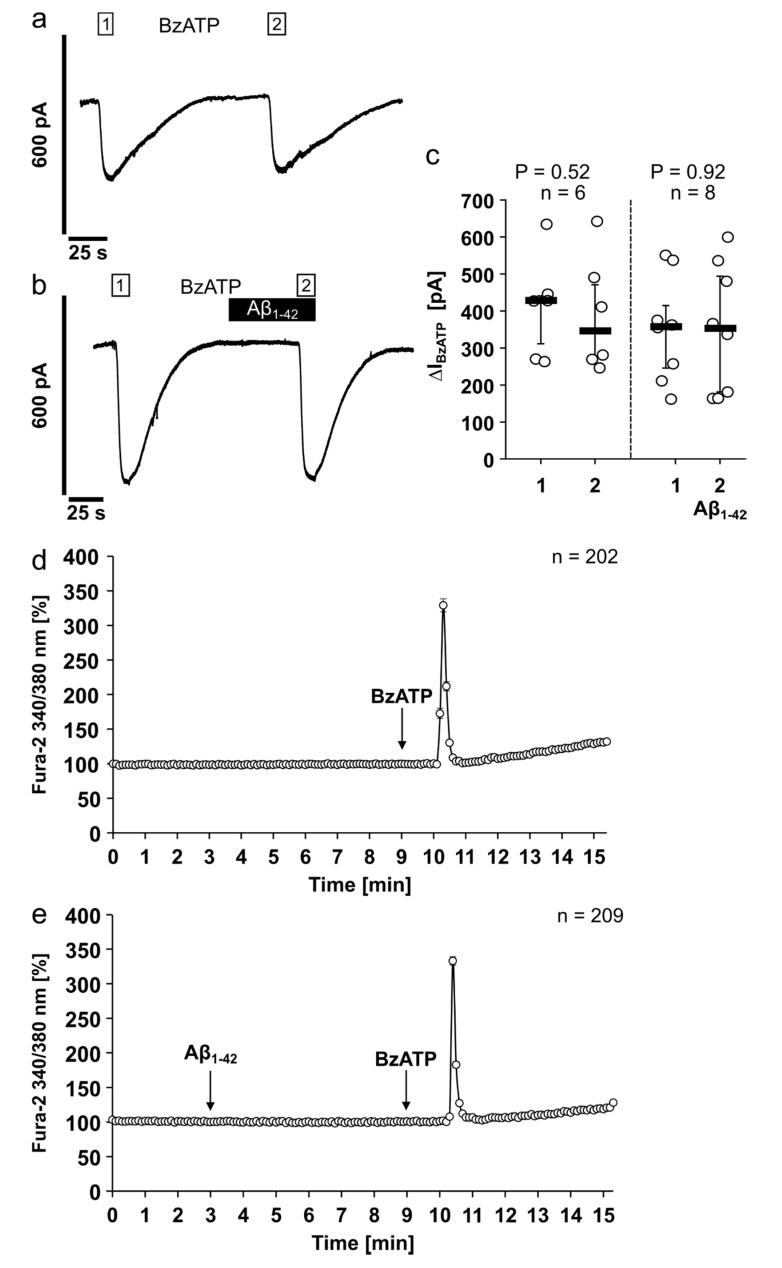
Amyloid beta peptide (Aβ_1-42_) has no impact on the ion channel activity of the human P2X7 receptor. (**a**–**c**) Whole-cell patch-clamp measurements were performed on HEK293 cells overexpressing human P2X7 receptor (HEK-P2X7R cells). (**a**,**b**) Depicted are representative current traces. (**a**,**c**) In control experiments, consecutive application of the P2X7 receptor agonist BzATP (100 µM, white bar) induced repetitive current stimulations (BzATP1 and 2). (**b**,**c**) After washout of the first BzATP stimulus, Aβ_1-42_ (5 µM, black bar) was applied and did not induce changes in ion currents. In the presence of Aβ_1-42_, BzATP induced repetitive current stimulations. (**c**) All BzATP-induced current changes (∆I_BzATP_) are shown as individual data points, bar represents median, whiskers encompass the 25th to 75th percentile. Statistical analyses were performed using the Friedman test followed by the Wilcoxon signed-rank test. (**d**,**e**) Calcium imaging experiments were performed on HEK-P2X7R cells. Intracellular calcium concentrations ([Ca^2+^]_i_) of HEK-P2X7R cells were recorded as Fura-2/AM (Fura-2) fluorescence intensity ratio of 340:380 nm excitation. For graphical representation and statistical analyses, signal intensities at time point 3 min were set to 100%, and all other values were calculated in accordance and are presented as mean ± SEM. (**a**) In control experiments, application of the P2X7 receptor agonist BzATP (100 µM) induced a rise in [Ca^2+^]_i_. (**b**) Aβ_1-42_ (5 µM) did not alter the [Ca^2+^]_i_. In the presence of Aβ_1-42_ additional application of BzATP induced a rise in [Ca^2+^]_i_. The BzATP-induced [Ca^2+^]_i_ changes (∆[Ca^2+^]_i_) in the absence ((**a**); ∆[Ca^2+^]_i_ = 239 ± 9%) and presence ((**b**); ∆[Ca^2+^]_i_ = 233 ± 6%) of Aβ_1-42_ were not significantly different (*p* = 0.92; Mann–Whitney U test).

**Figure 5 jcm-09-02887-f005:**
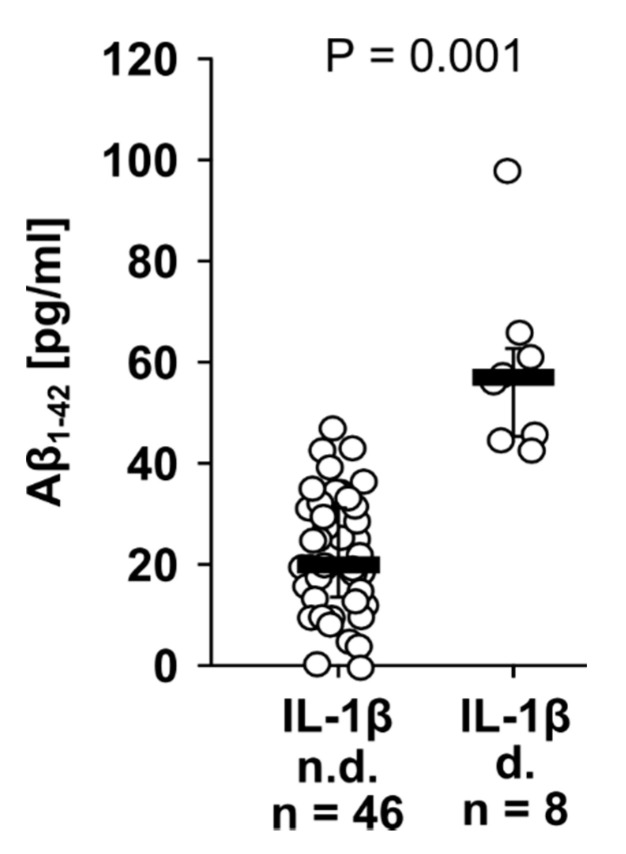
Interleukin-1β (IL-1β) is predominantly detected in the blood of surgical patients together with increased amyloid beta (Aβ_1-42_) levels. Blood was drawn from patients, who underwent major surgery, shortly before the operation (*n* = 12), 0 to 2 h (*n* = 10), 24 to 48 h (*n* = 12), 72 to 96 h (*n* = 12), and 7 to 9 days (*n* = 8) after the operation. Aβ_1-42_ and IL-1β were measured by ELISA in patient blood plasma. Aβ_1-42_ values of patients, in whom IL-1β levels were below the threshold of detection (n.d.), were compared to patients with detectable IL-1β (d). Data obtained for all time points were investigated collectively and are shown as individual data points, bar represents median, whiskers encompass the 25th to 75th percentile. Statistical analysis was performed using the Mann–Whitney U test.

**Figure 6 jcm-09-02887-f006:**
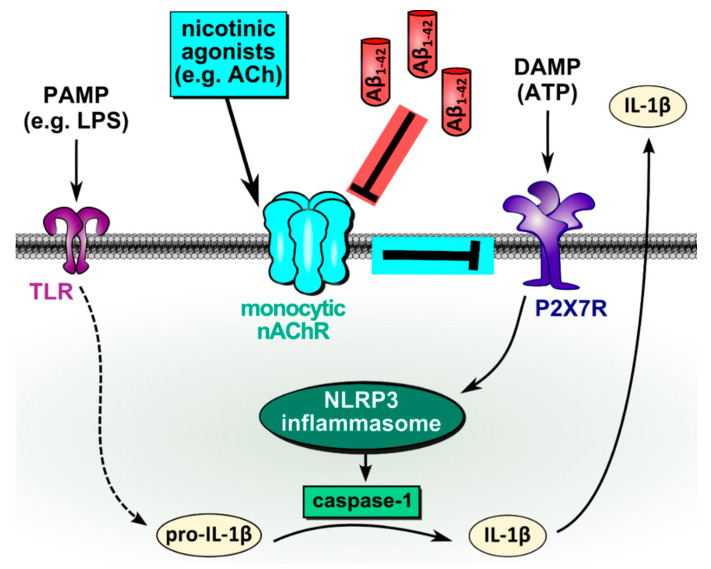
Schematic summary of the proposed mechanism. In LPS-primed monocytic cells, extracellular ATP, originating from dead or injured cells, binds to the ATP-sensitive P2X7R, induces NLRP3 inflammasome assembly, activation of caspase-1, cleavage of pro-IL-1β and release of bioactive IL-1β. This damage-associated release of IL-1β is inhibited by the activation of monocytic nAChRs containing subunits α9, α7, and/or α10. As shown previously, activation of nAChRs inhibits the ion channel function of P2X7R and, thus, ATP-induced IL-1β release [6,7]. Our data suggest that amyloid beta (Aβ_1-42_) antagonizes this inhibitory effect and enables monocytic release of IL-1β in the presence of nicotinic agonists. ACh, acetylcholine; ATP, adenosine triphosphate; DAMP, danger-associated molecular pattern; IL-1β, interleukin-1β; LPS, lipopolysaccharide; nAChR, nicotinic acetylcholine receptor; NLRP3, NACHT, LRR, and PYD domains-containing protein 3; P2X7R, P2X7 receptor; PAMP, pathogen-associated molecular pattern; TLR, Toll-like receptors.

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
