# Peer review of "Amyloid Beta Peptide (Aβ1-42) Reverses the Cholinergic Control of Monocytic IL-1β Release"

_jcm, 2020, doi:10.3390/jcm9092887_

Round 1

Reviewer 1 Report

This is potentially a very interesting continuation of the research by these authors showing nicotinic receptor function on monocytes.  In this paper, they claim evidence that the a-Beta peptide (1-42) associated with Alzheimer’s disease potentiates the release of IL-1Beta, a pro-inflammatory cytokine implicated in some of the pathological effects of Alzheimer’s.  If true, this suggests potential therapeutic approaches.

However, the references are distracting. For instance, in lines 100-103 the authors talk about pentraxin C-reactive protein complexed with phosphocholine as an agonist that they described in reference 37, but 37 is a 1996 paper by Roger Papke about choline being an agonist at alpha7 receptors,  This makes understanding the paper very difficult.  Later (Lines 109, 111) they refer to their work in references 37 and 38, but I think they mean references 30 and 31?  Also, they do not refer to older literature about a-Beta effects on nicotinic receptors.  These effects are confusing since it is described as an agonist (Dinelly et al. J. Biol. Chem. 277: 25056–25061, 2002), an antagonist (Liu et. al. PNAS 98: 4734–4739, 2001) or what can best be described as a partial agonist at alpha7 receptors (Dougherty et al. J. Neurosci., 23:6740–6747, 2003), just to name one receptors subtype. But it is important to include the earlier literature.

Line 456 “We and others demonstrated that leukocytic nAChRs do not function as classical ion channels, but rather exert metabotropic functions”.  That was the belief also of a7 nAChRs on macrophages until recently when Báez-Pagán et al. (J Neuroimmune Pharmacol (2015) 10:468–476 – which the authors should reference) showed ion channels in the presence of the positive allosteric modulator PNU-120596.  Have the authors tried positive allosteric modulators to try to boost channel activity in monocytes?

Author Response

This is potentially a very interesting continuation of the research by these authors showing nicotinic receptor function on monocytes.  In this paper, they claim evidence that the a-Beta peptide (1-42) associated with Alzheimer’s disease potentiates the release of IL-1Beta, a pro-inflammatory cytokine implicated in some of the pathological effects of Alzheimer’s.  If true, this suggests potential therapeutic approaches.

However, the references are distracting. For instance, in lines 100-103 the authors talk about pentraxin C-reactive protein complexed with phosphocholine as an agonist that they described in reference 37, but 37 is a 1996 paper by Roger Papke about choline being an agonist at alpha7 receptors,  This makes understanding the paper very difficult.  Later (Lines 109, 111) they refer to their work in references 37 and 38, but I think they mean references 30 and 31?

Response to the Reviewer: Thank you very much for drawing our attention to these mistakes. We sincerely apologize for the incorrect proofreading and the confusion this has created. We have now corrected the manuscript accordingly. All changes to the manuscript are highlighted using the “Track changes” function in Microsoft Word.

 Also, they do not refer to older literature about a-Beta effects on nicotinic receptors.  These effects are confusing since it is described as an agonist (Dinelly et al. J. Biol. Chem. 277: 25056–25061, 2002), an antagonist (Liu et. al. PNAS 98: 4734–4739, 2001) or what can best be described as a partial agonist at alpha7 receptors (Dougherty et al. J. Neurosci., 23:6740–6747, 2003), just to name one receptors subtype. But it is important to include the earlier literature.

Response to the Reviewer: Thank you very much for pointing this out. We have now included and discussed these references in the manuscript (line 506-511).

Line 456 “We and others demonstrated that leukocytic nAChRs do not function as classical ion channels, but rather exert metabotropic functions”.  That was the belief also of a7 nAChRs on macrophages until recently when Báez-Pagán et al. (J Neuroimmune Pharmacol (2015) 10:468–476 – which the authors should reference) showed ion channels in the presence of the positive allosteric modulator PNU-120596.  Have the authors tried positive allosteric modulators to try to boost channel activity in monocytes?

Response to the Reviewer: We gratefully appreciate your correction. We have now included this reference as well as a recently published paper from Siniavin et al. (Biomol. 2020, 10, doi:10.3390/biom10040507) in the manuscript (line 511-515). No, we have not tested positive allosteric modulators yet, but this is an interesting idea and is already on our agenda. We do not, however, feel that the results of these experiments should be included in the current study since this journal focuses on clinical medicine.

Reviewer 2 Report

In the manuscript entitled "Amyloid beta peptide enables monocytic IL1-beta release in the presence of nicotinic agonists" the authors present an interesting mechanism/interference in receptor-cytokine interaction in systemic imflammation.

The introduction leads well to the topic an the figures are sufficiently clear and structured. The discussion part gives a good overview of literature in the contex of the presented results.  The authors rigthly point out to the small cohort of patients and its limitation due to the discussed in vivo linkage. 

Author Response

In the manuscript entitled "Amyloid beta peptide enables monocytic IL1-beta release in the presence of nicotinic agonists" the authors present an interesting mechanism/interference in receptor-cytokine interaction in systemic imflammation.

The introduction leads well to the topic an the figures are sufficiently clear and structured. The discussion part gives a good overview of literature in the contex of the presented results.  The authors rigthly point out to the small cohort of patients and its limitation due to the discussed in vivo linkage. 

Response to the Reviewer: Thank you very much for the interest on the topic of our manuscript and for your kind comments.

Reviewer 3 Report

Richter et al describe in their manuscript the impact of Ab1-42 in the release of IL-1b from monocytic cells, in particular its counter-regulatory role of cholinergic mechanisms that had been described earlier by this group. The manuscript is well written and the presented new findings are certainly of interest. However, the data do not fully support the authors conclusion and the manuscripts needs major revision in order to be eligible for publication in the Journal of Clinical Medicine.

Major points:

1) The manuscript does not address the role of Ab1-42 as pro-inflammatory mediator. In the murine system it has been demonstrated that Ab1-42 can boost the transcription and expression of the pro-form of IL-1b in astrocytes (see Ebrahimi et al 2018 J Neuroinflam). Similarly, Ab1-42 has been shown to induce pro-IL-1bexpression in microglia and simultaneously serving as DAMP to trigger NLRP3 inflammasome activation and release of mature IL-1b (Halle et al 2008, Nature Immunology). Although these were murine cells the authors should address the possibility that Ab1-42 could simply boost Il1b expression rather than counter-regulating the inhibitory impact of cholinergic mechanisms. This could be achieved by measuring Il1b mRNA in cells stimulated in the presence of Ab1-42 and total intracellular IL-1b after LPS +/–Ab1-42 stimulation e.g. via Western Blot.

2) As the authors state in the conclusion: the manuscript does not provide any evidence for the interaction of Ab1-42 and nAChR. Since this interaction is the key of the concept presented here, the authors should make every effort to provide data indicating such an interaction. I agree with the authors that this might be challenging, however, one could e.g. perform another patch-clamp experiment similarly as in Hecker et al J Immunol 2015 (Fig.4B): since nicotine abrogates BzATP mediated P2X7 activation, the presence of Ab1-42 should reverse this effect, like mecamylamine did (Hecker et al J Immunol 2015 Fig.4C).

3) The observation that patients with detectable serum level of IL-1b have higher Ab1-42 serum level as patients where no IL-1b is detectable in the serum is surely interesting but it could be also “caused” by a more sever inflammatory reaction. The authors could provide an additional data analysis that correlates IL-1b level with Ab1-42 level. If higher Ab1-42 level correlate go hand in hand with lower IL-1b level this would strengthen the authors claim.  

Minor points:

  1. The title “Amyloid beta peptide (Aβ1–42) enables monocytic IL-1β release in the presence of nicotinic agonists” is misleading. It suggests that Aβ1–42 together with nicotinic agonists can induce IL-1b release, whereas the key message of the manuscript is that Aβ1–42 reverses the counter-regulatory action of nAChR in P2X7-mediated IL-1b release.
  2. [IL-1b %] for human PBMC needs to be changed to absolute values. Absolute values should be provided, even if the variation between donors is quite prominent
  3. Figure 4 and Figure 5 should be combined since they show highly related experiments pointing towards the same direction.
  4. Showing individual data points in the graphs is much appreciated, but the authors should check if the “n” matches the number of dots shown in the graph, e.g. this is not the case in Fig.1: groups 4 and 5 are n=5 and n=7 and not n=6. Same for Fig.2 group 3 (n=4)

Author Response

Richter et al describe in their manuscript the impact of Ab1-42 in the release of IL-1b from monocytic cells, in particular its counter-regulatory role of cholinergic mechanisms that had been described earlier by this group. The manuscript is well written and the presented new findings are certainly of interest. However, the data do not fully support the authors conclusion and the manuscripts needs major revision in order to be eligible for publication in the Journal of Clinical Medicine.

Response to the Reviewer: We appreciate the reviewer's interest on the topic of our manuscript and are grateful for the valuable suggestions that have helped improve this manuscript. All changes to the manuscript are highlighted by using the “Track changes” function in Microsoft Word.

Major points:

1) The manuscript does not address the role of Ab1-42 as pro-inflammatory mediator. In the murine system it has been demonstrated that Ab1-42 can boost the transcription and expression of the pro-form of IL-1b in astrocytes (see Ebrahimi et al 2018 J Neuroinflam). Similarly, Ab1-42 has been shown to induce pro-IL-1bexpression in microglia and simultaneously serving as DAMP to trigger NLRP3 inflammasome activation and release of mature IL-1b (Halle et al 2008, Nature Immunology). Although these were murine cells the authors should address the possibility that Ab1-42 could simply boost Il1b expression rather than counter-regulating the inhibitory impact of cholinergic mechanisms. This could be achieved by measuring Il1b mRNA in cells stimulated in the presence of Ab1-42 and total intracellular IL-1b after LPS +/–Ab1-42 stimulation e.g. via Western Blot.

Response to the Reviewer: Thank you very much for mentioning these publications. We have now included and discussed them in the manuscript (line 519 -532). However, we have already provided some evidence that Aβ1-42 does not simply boost the IL-1β expression. As Aβ1-42 is added to the cells immediately before applying BzATP and the release of IL-1β is measured already 30 min thereafter. It is unlikely that a relevant upregulation of pro-IL-1β expression occurs. In addition, and more importantly, we show in Figure 3b that when BzATP and Aβ1-42 are applied in the absence of any cholinergic agonist, no over-shooting IL-1β release is observed. This aspect is now explicitly discussed (line 522-525; line 531-532).

2) As the authors state in the conclusion: the manuscript does not provide any evidence for the interaction of Ab1-42 and nAChR. Since this interaction is the key of the concept presented here, the authors should make every effort to provide data indicating such an interaction. I agree with the authors that this might be challenging, however, one could e.g. perform another patch-clamp experiment similarly as in Hecker et al J Immunol 2015 (Fig.4B): since nicotine abrogates BzATP mediated P2X7 activation, the presence of Ab1-42 should reverse this effect, like mecamylamine did (Hecker et al J Immunol 2015 Fig.4C).

Response to the Reviewer: We fully agree with you that this experiment would be beneficial. However, as we were asked by the Journal to re-submit a revised version of our manuscript within 7 days, we were obviously not able to perform the experiments in due time. Furthermore, we feel that the results of these experiments would not be of interest for the readers of JCM as the journal has a focus on clinical medicine. Moreover, patch-clamp experiments would at best(again) indicate an interaction of Aβ1-42 and nAChR as shown by multiple groups previously. (See for example references [60] Wang et al. (J. Neurochem. 2000, 75, 1155–1161, doi:10.1046/j.1471-4159.2000.0751155.x), [66] Pettit et al. (J. Neurosci. 2001, 21, RC120) and [15] Parri et al. (Curr. Alzheimer Res. 2010, 7, 27–39, doi:10.2174/156720510790274464). More research and other techniques are needed to prove a direct interaction between Aβ1-42 and nAChR. Such a follow up study is already planned.

3) The observation that patients with detectable serum level of IL-1b have higher Ab1-42 serum level as patients where no IL-1b is detectable in the serum is surely interesting but it could be also “caused” by a more sever inflammatory reaction. The authors could provide an additional data analysis that correlates IL-1b level with Ab1-42 level. If higher Ab1-42 level correlate go hand in hand with lower IL-1b level this would strengthen the authors claim.  

Response to the Reviewer: Thank you very much for pointing this out. We have performed a correlation analysis and, indeed, have found a positive correlation (P = 0.0061). Please find the corresponding graph in the attachment. You will see that this graph is confusing, which is most probably due to the fact that the proportion of patients with detectable IL-1β is relatively low. Therefore, we decided to show Aβ1-42 values of patients, in whom IL-1β levels were below the threshold of detection compared to patients with detectable IL-1β. As stated in the manuscript “these data should be interpreted with utmost caution and further careful clinical research is warranted, because we only investigated a small cohort of patients, pooled data from blood collections at different perioperative time points and do not prove causality at all”(line 540-542), and “However, more studies including experimental in vivo studies and extended carefully controlled clinical studies are warranted to estimate the clinical relevance of our results” (line 561-562).

Minor points:

  1. The title “Amyloid beta peptide (Aβ1–42) enables monocytic IL-1β release in the presence of nicotinic agonists” is misleading. It suggests that Aβ1–42 together with nicotinic agonists can induce IL-1b release, whereas the key message of the manuscript is that Aβ1–42 reverses the counter-regulatory action of nAChR in P2X7-mediated IL-1b release.

Response to the Reviewer: Thank you very much for this comment. We have re-phrased the title to “Amyloid beta peptide (Aβ1–42) reverses the cholinergic control of monocytic IL-1β release”.

  1. [IL-1b %] for human PBMC needs to be changed to absolute values. Absolute values should be provided, even if the variation between donors is quite prominent

Response to the Reviewer: Thank you very much for pointing this out. Because of the variations between the donors, we analyzed the IL-1β vales of human PBMCs in % and we agree that the absolute values should be provided. The absolute values of IL-1β [pg/ml] are now included in the supplements (supplemental Table S2).

  1. Figure 4 and Figure 5 should be combined since they show highly related experiments pointing towards the same direction.

Response to the Reviewer: We have combined Figure 4 and 5 into a new Figure 4.

  1. Showing individual data points in the graphs is much appreciated, but the authors should check if the “n” matches the number of dots shown in the graph, e.g. this is not the case in Fig.1: groups 4 and 5 are n=5 and n=7 and not n=6. Same for Fig.2 group 3 (n=4)

Response to the Reviewer: Thank you very much for pointing out these errors. We have checked the data. The n-numbers in the figures were correct, but it seems that some conversion failures have occurred with superimposed data points. We sincerely apologize for the incorrect proofreading. We have corrected Figure 1 and Figure 2 in the revised version of the manuscript.

Reviewer 4 Report

In this manuscript " Amyloid beta-peptide (Aβ1–42) enables monocytic IL-1β release in the presence of nicotinic agonists” Katrin Richter et.al. made an effort to test the hypothesis that Amyloid-β peptide (Aβ1-42) reverses the inhibitory effect of nicotinic agonists on ATP mediated release of IL-1β by human monocytic cells.

The comments and suggestions for this manuscript are as follows-

  1. Since this journal is specialized for clinical medicine, the authors should provide a more comprehensive introduction, discussion, and conclusion with a clinical perspective.
  2. Page 4, section 2.4. What are the clinical history of patients and the reason for the major surgery? Are they suffering from sepsis condition? These patients must be on the medication if they are undergoing major surgery. Are the prescribed medicines affecting the release of IL-1β? The author should explain these issues and add details in the manuscripts, preferably in the table format.
  3. The experimental design and result analysis sections are quite impressive for the in vitro experiments.
  4. Figures are concise including proper statistical analysis, wherever required.

Author Response

In this manuscript " Amyloid beta-peptide (Aβ1–42) enables monocytic IL-1β release in the presence of nicotinic agonists” Katrin Richter et.al. made an effort to test the hypothesis that Amyloid-β peptide (Aβ1-42) reverses the inhibitory effect of nicotinic agonists on ATP mediated release of IL-1β by human monocytic cells.

The comments and suggestions for this manuscript are as follows-

1. Since this journal is specialized for clinical medicine, the authors should provide a more comprehensive introduction, discussion, and conclusion with a clinical perspective.

Response to the Reviewer: Thank you very much for pointing this out. We amended both, introduction and discussion, and included a more clinical perspective, for example see line 558-562. All changes in the manuscript are highlighted by using the “Track changes” function in Microsoft Word. Yet, we are very careful regarding possible clinical applications of our findings, since more studies, especially in vivo are warranted.

2. Page 4, section 2.4. What are the clinical history of patients and the reason for the major surgery? Are they suffering from sepsis condition? These patients must be on the medication if they are undergoing major surgery. Are the prescribed medicines affecting the release of IL-1β? The author should explain these issues and add details in the manuscripts, preferably in the table format.

Response to the Reviewer: Thank you very much. We agree with you and have included the following information concerning the clinical history of the patients in the revised version of the manuscript (line 198-203): “All patients underwent elective surgery for oncological tumor resection of the esophagus (n = 8) or the pancreas (n = 4). Patients with preexisting increased inflammatory parameters and septic patients were excluded from the study. The medication prescribed prior to surgery was checked for potential interactions with IL-1β, as far as known from the literature. All patients underwent anesthesia according to a standardized regimen”.

3. The experimental design and result analysis sections are quite impressive for the in vitro experiments. Figures are concise including proper statistical analysis, wherever required.

Response to the Reviewer: Thank you very much for your kind remarks and we are grateful for the valuable suggestions that have helped to improve this manuscript.

Round 2

Reviewer 1 Report

This is better.  At least the references correspond to the text.

Author Response

Again, thank you very much for valuable suggestions that helped to improve this manuscript.

Reviewer 3 Report

Response to the Reviewer: We appreciate the reviewer's interest on the topic of our manuscript and are grateful for the valuable suggestions that have helped improve this manuscript. All changes to the manuscript are highlighted by using the “Track changes” function in Microsoft Word.

Major points:

1) The manuscript does not address the role of Ab1-42 as pro-inflammatory mediator. In the murine system it has been demonstrated that Ab1-42 can boost the transcription and expression of the pro-form of IL-1b in astrocytes (see Ebrahimi et al 2018 J Neuroinflam). Similarly, Ab1-42 has been shown to induce pro-IL-1bexpression in microglia and simultaneously serving as DAMP to trigger NLRP3 inflammasome activation and release of mature IL-1b (Halle et al 2008, Nature Immunology). Although these were murine cells the authors should address the possibility that Ab1-42 could simply boost Il1b expression rather than counter-regulating the inhibitory impact of cholinergic mechanisms. This could be achieved by measuring Il1b mRNA in cells stimulated in the presence of Ab1-42 and total intracellular IL-1b after LPS +/–Ab1-42 stimulation e.g. via Western Blot.

Response to the Reviewer: Thank you very much for mentioning these publications. We have now included and discussed them in the manuscript (line 519 -532). However, we have already provided some evidence that Aβ1-42 does not simply boost the IL-1β expression. As Aβ1-42 is added to the cells immediately before applying BzATP and the release of IL-1β is measured already 30 min thereafter. It is unlikely that a relevant upregulation of pro-IL-1β expression occurs. In addition, and more importantly, we show in Figure 3b that when BzATP and Aβ1-42 are applied in the absence of any cholinergic agonist, no over-shooting IL-1β release is observed. This aspect is now explicitly discussed (line 522-525; line 531-532).

Response to the authors:
In Fig.3B one can actually see that the addition of
Ab1-42 has the capacity to alter IL1b release in both directions in the absence of cholinergic agonists: 3/7 individuals show a stronger release of Il1b and 4/7 individuals show less Il1b release. Though there is no difference in the means of BzATP vs BzATP + Ab1-42 regarding IL1b release, there is certainly an impact of Ab1-42 on the individual level. If the authors wish not to perform additional Il1b qPCR or IL1b WB they should at least measure the IL1b release from U937 cells under the conditions LPS/BzATP vs LPS/BzATP/Ab1-42 in the absence of any cholinergic agonist.

2) As the authors state in the conclusion: the manuscript does not provide any evidence for the interaction of Ab1-42 and nAChR. Since this interaction is the key of the concept presented here, the authors should make every effort to provide data indicating such an interaction. I agree with the authors that this might be challenging, however, one could e.g. perform another patch-clamp experiment similarly as in Hecker et al J Immunol 2015 (Fig.4B): since nicotine abrogates BzATP mediated P2X7 activation, the presence of Ab1-42 should reverse this effect, like mecamylamine did (Hecker et al J Immunol 2015 Fig.4C).

Response to the Reviewer: We fully agree with you that this experiment would be beneficial. However, as we were asked by the Journal to re-submit a revised version of our manuscript within 7 days, we were obviously not able to perform the experiments in due time. Furthermore, we feel that the results of these experiments would not be of interest for the readers of JCM as the journal has a focus on clinical medicine. Moreover, patch-clamp experiments would at best(again) indicate an interaction of Aβ1-42 and nAChR as shown by multiple groups previously. (See for example references [60] Wang et al. (J. Neurochem. 2000, 75, 1155–1161, doi:10.1046/j.1471-4159.2000.0751155.x), [66] Pettit et al. (J. Neurosci. 2001, 21, RC120) and [15] Parri et al. (Curr. Alzheimer Res. 2010, 7, 27–39, doi:10.2174/156720510790274464). More research and other techniques are needed to prove a direct interaction between Aβ1-42 and nAChR. Such a follow up study is already planned.

Response to the authors:
The reviewer encourages the authors to ask the editorial office for more time to perform the patch clamp experiment (and the above mentioned assay with the U937 cells). If the authors can demonstrate that
Ab1-42 can reverse the nicotine-mediated dampening of BzATP-triggered P2X7 activation these results will significantly strengthen the authors claim. To make it more clear this time: performing this experiment is mandatory. Further, the reviewer is very certain that the esteemed readership of JCM will value thoroughly performed research.

3) The observation that patients with detectable serum level of IL-1b have higher Ab1-42 serum level as patients where no IL-1b is detectable in the serum is surely interesting but it could be also “caused” by a more sever inflammatory reaction. The authors could provide an additional data analysis that correlates IL-1b level with Ab1-42 level. If higher Ab1-42 level correlate go hand in hand with lower IL-1b level this would strengthen the authors claim. 

Response to the Reviewer: Thank you very much for pointing this out. We have performed a correlation analysis and, indeed, have found a positive correlation (P = 0.0061). Please find the corresponding graph in the attachment. You will see that this graph is confusing, which is most probably due to the fact that the proportion of patients with detectable IL-1β is relatively low. Therefore, we decided to show Aβ1-42 values of patients, in whom IL-1β levels were below the threshold of detection compared to patients with detectable IL-1β. As stated in the manuscript “these data should be interpreted with utmost caution and further careful clinical research is warranted, because we only investigated a small cohort of patients, pooled data from blood collections at different perioperative time points and do not prove causality at all”(line 540-542), and “However, more studies including experimental in vivo studies and extended carefully controlled clinical studies are warranted to estimate the clinical relevance of our results” (line 561-562).

Response to the authors:
The reviewer appreciates the authors response and he agrees that the overwhelming amount of patients without detectable IL1b in the serum might skew the interpretation. However, by eye, one can see an inverse correlation (high Ab1-42 and low IL1b) if one sets the focus on the 9 patients that have detectable level of IL-1b in the serum. This would be in line with the authors argumentation and, if the authors wish, could be added to Fig.5. Limitations could be discussed as stated by the authors in their response.

Minor points:

1.The title “Amyloid beta peptide (Aβ1–42) enables monocytic IL-1β release in the presence of nicotinic agonists” is misleading. It suggests that Aβ1–42 together with nicotinic agonists can induce IL-1b release, whereas the key message of the manuscript is that Aβ1–42 reverses the counter-regulatory action of nAChR in P2X7-mediated IL-1b release.

Response to the Reviewer: Thank you very much for this comment. We have re-phrased the title to “Amyloid beta peptide (Aβ1–42) reverses the cholinergic control of monocytic IL-1β release”.

2.[IL-1b %] for human PBMC needs to be changed to absolute values. Absolute values should be provided, even if the variation between donors is quite prominent

Response to the Reviewer: Thank you very much for pointing this out. Because of the variations between the donors, we analyzed the IL-1β vales of human PBMCs in % and we agree that the absolute values should be provided. The absolute values of IL-1β [pg/ml] are now included in the supplements (supplemental Table S2).

3. Figure 4 and Figure 5 should be combined since they show highly related experiments pointing towards the same direction.

Response to the Reviewer: We have combined Figure 4 and 5 into a new Figure 4. 

4.Showing individual data points in the graphs is much appreciated, but the authors should check if the “n” matches the number of dots shown in the graph, e.g. this is not the case in Fig.1: groups 4 and 5 are n=5 and n=7 and not n=6. Same for Fig.2 group 3 (n=4)

Response to the Reviewer: Thank you very much for pointing out these errors. We have checked the data. The n-numbers in the figures were correct, but it seems that some conversion failures have occurred with superimposed data points. We sincerely apologize for the incorrect proofreading. We have corrected Figure 1 and Figure 2 in the revised version of the manuscript.

Response to the authors:

All fine

Author Response

Response to the Reviewer: We appreciate the reviewer's interest on the topic of our manuscript and are grateful for the valuable suggestions that have helped improve this manuscript. All changes to the manuscript are highlighted by using the “Track changes” function in Microsoft Word.

Major points:

1) The manuscript does not address the role of Ab1-42 as pro-inflammatory mediator. In the murine system it has been demonstrated that Ab1-42 can boost the transcription and expression of the pro-form of IL-1b in astrocytes (see Ebrahimi et al 2018 J Neuroinflam). Similarly, Ab1-42 has been shown to induce pro-IL-1bexpression in microglia and simultaneously serving as DAMP to trigger NLRP3 inflammasome activation and release of mature IL-1b (Halle et al 2008, Nature Immunology). Although these were murine cells the authors should address the possibility that Ab1-42 could simply boost Il1b expression rather than counter-regulating the inhibitory impact of cholinergic mechanisms. This could be achieved by measuring Il1b mRNA in cells stimulated in the presence of Ab1-42 and total intracellular IL-1b after LPS +/–Ab1-42 stimulation e.g. via Western Blot.

Response to the Reviewer: Thank you very much for mentioning these publications. We have now included and discussed them in the manuscript (line 519 -532). However, we have already provided some evidence that Aβ1-42 does not simply boost the IL-1β expression. As Aβ1-42 is added to the cells immediately before applying BzATP and the release of IL-1β is measured already 30 min thereafter. It is unlikely that a relevant upregulation of pro-IL-1β expression occurs. In addition, and more importantly, we show in Figure 3b that when BzATP and Aβ1-42 are applied in the absence of any cholinergic agonist, no over-shooting IL-1β release is observed. This aspect is now explicitly discussed (line 522-525; line 531-532).

Response to the authors:
In Fig.3B one can actually see that the addition of
Ab1-42 has the capacity to alter IL1b release in both directions in the absence of cholinergic agonists: 3/7 individuals show a stronger release of Il1b and 4/7 individuals show less Il1b release. Though there is no difference in the means of BzATP vs BzATP + Ab1-42 regarding IL1b release, there is certainly an impact of Ab1-42 on the individual level. If the authors wish not to perform additional Il1b qPCR or IL1b WB they should at least measure the IL1b release from U937 cells under the conditions LPS/BzATP vs LPS/BzATP/Ab1-42 in the absence of any cholinergic agonist.

Response to the Reviewer Round 2: Thank you for your suggestions. We have shown that there is no statistical evidence for an effect of Aβ1-42 on the BzATP-induced IL-1β release by primary human PBMCs in absence of a cholinergic agonist (Figure 3b). From our point of view, data on primary PBMCs are more meaningful and we disagree that doing the same experiments again on the U937 cell line would improve our findings. We agree that Western blot or quantitative RT-PCR analysis might be an option, but these experiments will take some time. Moreover, Ebrahimi and colleagues have used a stimulation time with Aβ1-42 for 3 h to 6 h in murine astrocytes (J. Neuroinflamm 2018). In the present study, Aβ1-42 was applied very shortly (max. 3 min) before BzATP, and thus, it is very unlikely that pro-IL-1β expression levels are changed. This aspect is discussed in our manuscript (line 522-525; line 531-532).

2) As the authors state in the conclusion: the manuscript does not provide any evidence for the interaction of Ab1-42 and nAChR. Since this interaction is the key of the concept presented here, the authors should make every effort to provide data indicating such an interaction. I agree with the authors that this might be challenging, however, one could e.g. perform another patch-clamp experiment similarly as in Hecker et al J Immunol 2015 (Fig.4B): since nicotine abrogates BzATP mediated P2X7 activation, the presence of Ab1-42 should reverse this effect, like mecamylamine did (Hecker et al J Immunol 2015 Fig.4C).

Response to the Reviewer: We fully agree with you that this experiment would be beneficial. However, as we were asked by the Journal to re-submit a revised version of our manuscript within 7 days, we were obviously not able to perform the experiments in due time. Furthermore, we feel that the results of these experiments would not be of interest for the readers of JCM as the journal has a focus on clinical medicine. Moreover, patch-clamp experiments would at best(again) indicate an interaction of Aβ1-42 and nAChR as shown by multiple groups previously. (See for example references [60] Wang et al. (J. Neurochem. 2000, 75, 1155–1161, doi:10.1046/j.1471-4159.2000.0751155.x), [66] Pettit et al. (J. Neurosci. 2001, 21, RC120) and [15] Parri et al. (Curr. Alzheimer Res. 2010, 7, 27–39, doi:10.2174/156720510790274464 ) . More research and other techniques are needed to prove a direct interaction between Aβ1-42 and nAChR. Such a follow up study is already planned.

Response to the authors:
The reviewer encourages the authors to ask the editorial office for more time to perform the patch clamp experiment (and the above mentioned assay with the U937 cells). If the authors can demonstrate that
Ab1-42 can reverse the nicotine-mediated dampening of BzATP-triggered P2X7 activation these results will significantly strengthen the authors claim. To make it more clear this time: performing this experiment is mandatory. Further, the reviewer is very certain that the esteemed readership of JCM will value thoroughly performed research.

Response to the Reviewer Round 2: Thank you very much for the recommendation. We were obviously not able to perform patch clamp experiments due in time (within 3 days). Again, we disagree that whole-cell patch clamp experiments on U937 cells alone will help to prove a direct interaction of Aβ1-42 with nAChRs. Further detailed experiments would be required to do so, and this might take about 2 years.

3) The observation that patients with detectable serum level of IL-1b have higher Ab1-42 serum level as patients where no IL-1b is detectable in the serum is surely interesting but it could be also “caused” by a more sever inflammatory reaction. The authors could provide an additional data analysis that correlates IL-1b level with Ab1-42 level. If higher Ab1-42 level correlate go hand in hand with lower IL-1b level this would strengthen the authors claim. 

Response to the Reviewer: Thank you very much for pointing this out. We have performed a correlation analysis and, indeed, have found a positive correlation (P = 0.0061). Please find the corresponding graph in the attachment. You will see that this graph is confusing, which is most probably due to the fact that the proportion of patients with detectable IL-1β is relatively low. Therefore, we decided to show Aβ1-42 values of patients, in whom IL-1β levels were below the threshold of detection compared to patients with detectable IL-1β. As stated in the manuscript “these data should be interpreted with utmost caution and further careful clinical research is warranted, because we only investigated a small cohort of patients, pooled data from blood collections at different perioperative time points and do not prove causality at all”(line 540-542), and “However, more studies including experimental in vivo studies and extended carefully controlled clinical studies are warranted to estimate the clinical relevance of our results” (line 561-562).

Response to the authors:
The reviewer appreciates the authors response and he agrees that the overwhelming amount of patients without detectable IL1b in the serum might skew the interpretation. However, by eye, one can see an inverse correlation (high Ab1-42 and low IL1b) if one sets the focus on the 9 patients that have detectable level of IL-1b in the serum. This would be in line with the authors argumentation and, if the authors wish, could be added to Fig.5. Limitations could be discussed as stated by the authors in their response.

Response to the Reviewer Round 2: When we prepared the manuscript, performed the correlation analysis, and produced the corresponding figure, we also saw the inverse correlation among those patients, who had detectable IL-1beta levels. As this observation bases on only 9 patients, we decided to present the data the way we did in the first version of our manuscript. The only conclusion we can draw is, that patients with detectable IL-1β levels have significantly higher Aβ1-42 levels compared to those with undetectable IL-1β. We refrain from showing the correlation analysis to avoid confusion and unjustified conclusions.

Minor points:

1.The title “Amyloid beta peptide (Aβ1–42) enables monocytic IL-1β release in the presence of nicotinic agonists” is misleading. It suggests that Aβ1–42 together with nicotinic agonists can induce IL-1b release, whereas the key message of the manuscript is that Aβ1–42 reverses the counter-regulatory action of nAChR in P2X7-mediated IL-1b release.

Response to the Reviewer: Thank you very much for this comment. We have re-phrased the title to “Amyloid beta peptide (Aβ1–42) reverses the cholinergic control of monocytic IL-1β release”.

2.[IL-1b %] for human PBMC needs to be changed to absolute values. Absolute values should be provided, even if the variation between donors is quite prominent

Response to the Reviewer: Thank you very much for pointing this out. Because of the variations between the donors, we analyzed the IL-1β vales of human PBMCs in % and we agree that the absolute values should be provided. The absolute values of IL-1β [pg/ml] are now included in the supplements (supplemental Table S2).

  1. Figure 4 and Figure 5 should be combined since they show highly related experiments pointing towards the same direction.

Response to the Reviewer: We have combined Figure 4 and 5 into a new Figure 4. 

4.Showing individual data points in the graphs is much appreciated, but the authors should check if the “n” matches the number of dots shown in the graph, e.g. this is not the case in Fig.1: groups 4 and 5 are n=5 and n=7 and not n=6. Same for Fig.2 group 3 (n=4)

Response to the Reviewer: Thank you very much for pointing out these errors. We have checked the data. The n-numbers in the figures were correct, but it seems that some conversion failures have occurred with superimposed data points. We sincerely apologize for the incorrect proofreading. We have corrected Figure 1 and Figure 2 in the revised version of the manuscript.

Response to the authors:

All fine

Response to the Reviewer Round 2: Again, thank you very much for the valuable suggestions that have helped to improve this manuscript.

Reviewer 4 Report

The author's responses are satisfactory.

Author Response

Again, thank you very much for valuable suggestions that have helped to improve this manuscript.